# Isolation and Production of Human Monoclonal Antibody Proteins against a *Toxocara canis* Excretory–Secretory Recombinant Antigen

**DOI:** 10.3390/pathogens11111232

**Published:** 2022-10-25

**Authors:** Zamrina Baharudeen, Rahmah Noordin, Lim Theam Soon, Dinesh Balachandra, Nor Suhada Anuar, Fatin Hamimi Mustafa, Anizah Rahumatullah

**Affiliations:** Institute for Research in Molecular Medicine (INFORMM), Universiti Sains Malaysia, Penang 11800, Malaysia

**Keywords:** *Toxocara canis*, phage display technology, recombinant monoclonal antibodies, antigen-antibody binding assays, antigen detection ELISA

## Abstract

Toxocariasis is a widespread zoonotic parasitic disease with a significant socioeconomic impact, particularly on underprivileged communities. Limitations of existing diagnostic tools and vague presenting symptoms may lead to misdiagnosis, thus underestimating the actual global impact of the disease. The present study describes the isolation and production of novel recombinant monoclonal antibodies against *Toxocara canis* recombinant TES-26 antigen (rTES-26) utilizing a human helminth scFv phage display library. The isolated antibody clones were characterized based on their gene sequences and binding characteristics. Three clones representing unique gene families (clone 48: IgHV3-LV1; clone 49: IgHV3-LV3; clone 50: IgHV6-LV3) were isolated, but only clones 48 and 49 showed successful insertion of the full-length scFv antibody sequence after sub-cloning. Both clones produced antibody proteins of good solubility and satisfactory yield and purity. Binding assays via Western blot and ELISA using rTES-26 and *Toxocara canis* native protein showed that both monoclonal antibodies were highly specific and sensitive to the target antigen. A preliminary antigen detection ELISA showed the diagnostic potential of the monoclonal antibody proteins. The proteins can also be useful in studying host–parasite interactions and therapeutic applications.

## 1. Introduction

Human toxocariasis is a zoonotic parasitic infection with a worldwide prevalence. It is mainly caused by the larvae of *Toxocara canis (T. canis)* from canids and also by *Toxocara cati* (*T. cati)* and *Toxocara malaysiensis* (*T. malaysiensis)* from felids [1]. Toxocariasis is a silent public health issue with significant socioeconomic impact on underprivileged communities. It has been listed by the Centers for Disease Control (CDC), USA, as one of the five most neglected diseases [2].

The global seroprevalence (anti-*Toxocara* spp. serum antibodies) was estimated to be 19% and varies depending on country and region [3]. It is particularly prevalent in tropical and sub-tropical countries with limited dog treatment and population control [4]. Differences in the sensitivity and specificity among the serological assays used in the prevalence studies have also contributed to the observed data variations [3]. 

Human infection occurs through accidental ingestion of infective eggs from contaminated soil, water, food, or utensils [5], direct contact with infected pets, or eating raw or undercooked meat or organs containing encapsulated larvae from paratenic hosts [2,6]. After ingestion, the larvae in the ova are released in the intestine, penetrate the mucosa and migrate to different sites such eyes, liver, lungs, and central nervous system. The immune responses to the migrating larvae cause local inflammation, eosinophilia, increased cytokines production and specific antibodies [5].

Clinical forms of toxocariasis include visceral larval migrans (VLM), covert or common toxocariasis (CT), ocular larval migrans (OLM), and neurotoxocariasis (NT) [5]. The clinical diagnosis of *Toxocara* spp. infection can be challenging as many infected patients are asymptomatic or have inapparent symptoms [7]. Furthermore, there are some limitations of available diagnostic tools [1] and a lack of awareness concerning human toxocariasis among medical practitioners.

Diagnosis of human toxocariasis is primarily based on clinical, epidemiological, and serological detection methods [8]. Serology, mainly by enzyme-linked immunosorbent assay (ELISA), has been a common method for disease detection, mapping seroprevalence, and epidemiological studies. Most commercially available ELISA kits use *T. canis* excretory–secretory (TES) antigens from the second-stage larvae culture [9]. Other reported detection platforms are Western blot, which can confirm the ELISA results, lateral flow rapid test, and immunosensor [8]. However, the native TES antigen production is laborious, time-consuming, and low yield. 

In recent years, the serodiagnosis of toxocariasis has been improved by using recombinant forms of TES antigens to replace the native antigens. Recombinant TES antigens provide a significant detection advantage due to their infinite production capacity and enhanced diagnostic sensitivity and specificity [10]. Several recombinant TES proteins have shown good diagnostic value, i.e., rTES-26, rTES-30, rTES-120, and *T. cati* rTES-120 [10,11,12,13]. 

Most of the current serological detection systems are based on antibody detection, and the approach may cause difficulty in discriminating between past exposure and active infection [14]. Alternatively, an antigen detection assay could detect circulating *Toxocara* spp. antigens and diagnose an active infection, thus addressing the challenges related to antibody assays. An antigen detection assay is conceivable since the *Toxocara* spp. larvae in the human body are thought to be dormant (hypobiosis state) and remain viable for several years, thus may secrete antigens [15]. Antigen detection-based serology assay utilizes an antibody (polyclonal or monoclonal) that binds to the parasite’s circulating antigen. There are reports on polyclonal-based antigen detection ELISAs for toxocariasis; however, they showed low diagnostic sensitivity and cross-reactivity problems [16,17]. Meanwhile, monoclonal antibody-based antigen detection assays have shown increased specificity using serum samples from humans with polyparatism [18,19].

Besides ELISA, monoclonal antibodies can be applied in other diagnostic platforms to detect circulating antigens [20]. Recombinant monoclonal antibodies against *Toxocara* spp. are promising for developing such assays and can also be used as quality control reagents for commercial kits [8]. The monoclonal antibodies can also facilitate the analysis of host–parasite interactions and the finding of antigens that induce protective responses in the immunized host. To date, there is no commercial antigen detection test for toxocariasis. Thus, the present study aimed to isolate novel monoclonal antibody proteins against the *T. canis* rTES-26 antigen. We selected recombinant rTES-26 as the target antigen since previous studies have shown it to be highly specific and sensitive in diagnosing human toxocariasis [10,11,13,21]. 

## 2. Materials and Methods

### 2.1. Recombinant TES-26 Protein Preparation

The expression and purification of rTES-26 were performed as previously reported [13]. Briefly, the protein was purified using nickel–nitrilotriacetic acid (Ni–NTA) resin (complete His-Tag purification resin, Roche, Germany). The purified protein was separated using a 10% SDS-PAGE gel, and verified by Western blot using 1:1000 dilution of anti-6x His antibody conjugated to horseradish peroxidase (HRP) (Novagen Sigma-Aldrich, Darmstadt, Germany). The result was then developed using Super Signal West Pico PLUS chemiluminescent substrate (Thermo Scientific, St. Peters, CA, USA) on CL-XPosureTM films (Thermo Scientific, Waltham, MA, USA). The concentration of the purified protein was determined by absorbance measurements at OD750 using a spectrophotometer (Thermo Spectronic, Walthman, MA, USA).

### 2.2. Recombinant TES-26 Protein Sequence Analysis

Basic Local Alignment Search Tool (BLAST) analysis was performed using rTES-26 antigen sequence against *Brugia malayi* sequences to determine the extent of sequence identities at gene and protein levels. 

### 2.3. Biopanning, Phage ELISA and DNA Sequencing

Biopanning was performed using a human helminth scFv phage display library against the rTES-26 [22]. The library was previously used to isolate scFv clones against other target antigens [22,23,24,25]. In brief, three rounds of biopanning were carried out using 50 µg/mL rTES-26 protein and polyclonal phage ELISA was performed at the end of the process to obtain the rTES-26-specific polyclonal antibody enrichments. The positive clones were analyzed using monoclonal phage ELISA as previously described [22]. Subsequently, the positive antibody clones from the monoclonal ELISA were grown at 37 °C, 200 rpm overnight (~16 h). The cell pellets were then harvested, and plasmids were purified using the QIAprep Spin Miniprep Kit (Qiagen, Hilden, Germany). The plasmids were sent for sequencing (FIRST Base Laboratories, Malaysia), and the results were analyzed using the IMGT/V-QUEST bioinformatics tool available at the International ImMunoGeneTics information system^®^ or IMGT^®^ [26,27]. 

### 2.4. Recombinant Monoclonal Antibody Protein Expression and Purification

In order to obtain soluble scFv expression with improved yield and purity, plasmids of positive clones that exhibited complete gene sequences were subcloned into pET-51 (b) + vectors (fused with Strep-Tag and with C-terminal His-tag), then transformed into SHuffle^®^ T7 Express Competent *Escherichia coli* cells (NEB, Ipswich, MA, USA).

A starter culture was prepared by inoculating a single colony from the transformed plate into 10 mL of 2-YT broth supplemented with 100 µg/mL ampicillin with 2% glucose at 37 °C, 200 rpm, overnight. The following day, 10 mL of the culture was inoculated into 1 L of 2-YT medium supplemented with 100 µg/mL ampicillin and 0.2% glucose, and the culture was grown at 37 °C, 200 rpm until the OD_600 nm_ reached 0.6 to 0.7. Protein expression was induced with 1 mM IPTG and cultured for 16 h at 25 °C, 200 rpm, then harvested by centrifugation at 10,000× *g* for 10 min at 4 °C. The pellet was resuspended in cold lysis buffer (50 mM NaH_2_PO_4_, 500 mM NaCl, 10 mM) containing imidazole, 0.5 mg/mL lysozyme, and protease inhibitors. The mixture was incubated at 4 °C on a boule mixer for 30 min and sonicated (Qsonica, Melville, NY, USA) for 2 min with 30 s “on” and 10 s “off” cycles at 4.0 Hz output. The disrupted cells were centrifuged at 10,000× *g* at 4 °C for 30 min. Then, 0.5 µg/mL DNase 1 was added to the supernatant and incubated at 4 °C for 15 min, followed by another round of centrifugation at 10,000× *g* at 4°C for 30 min. 

The final supernatant was filtered through a 0.45 µm filter and purified using a purification column containing nitrilotriacetic acid (Ni–NTA) (Qiagen GmbH, Hilden, Germany), according to the manufacturer’s instructions. The purity of the protein fractions was verified by SDS PAGE. Western blotting was performed using HRP-conjugated anti-His and StrepTactin antibodies (BioRad, California, CA, USA) (1:5000), then developed on films using SuperSignal West Pico PLUS chemiluminescent substrate (Thermo Scientific, St. Peters, MO, USA).

### 2.5. Antigen-Antibody Binding Assays

#### 2.5.1. Western Blot Using Recombinant and Native Antigen Proteins

Recombinant antigen–antibody Western blot was performed by running 20 µg of rTES-26 antigen on 10% SDS-PAGE at a constant voltage of 100 V for 1 h. The protein was transferred onto a nitrocellulose membrane using a Trans-Blot Semi-Dry system (BioRad, Hercules, CA, USA) at a constant voltage of 12 V for 30 min. The membrane was then cut into strips and blocked with MTBST, i.e., 5% skim milk in Tris-Buffered Saline with 0.1% Tween 20, pH 7.6 (TBST) for 1 h, followed by three washes at 5-min intervals with TBST. After washing, the strips were incubated with 0.5 mg/mL of the respective recombinant monoclonal antibody (rmAb) proteins at 4 °C overnight. The next day, the strips were rewashed and incubated with StrepTactin-HRP at 1:5000 dilution in TBST for 1 h at room temperature, then developed on film using SuperSignal West Pico PLUS chemiluminescent substrate.

Native antigen–antibody Western blot was performed similarly as described above but with slight modifications. i.e., the amount of protein loaded on SDS-PAGE was 50 µg of *T. canis* lysate protein, and StrepTactin–HRP was diluted at 1:3000 and 1:10,000. Previously produced rabbit polyclonal anti-rTES-26 was used as the positive control and detected using 1:10,000 dilution of goat anti-rabbit IgG-HRP conjugate (Bio-rad Laboratories, Hercules, CA, USA).

#### 2.5.2. Immunoassays Using Recombinant and Native Antigen Proteins

Recombinant antigen–antibody ELISA was performed by coating 50 μg/mL of rTES-26 on a Maxisorb ELISA microtiter plate (Nunc, Rochester, NY, USA) with carbonate buffer (pH 9.6) at 4 °C overnight. The next day, the microtiter plate was washed three times at 5 min intervals with PBST at 800 rpm on a plate shaker to remove unbound antigen, then blocked with MPBST for 1 h at 37 °C, 600 rpm. The microtiter plate was washed, and the wells were separately incubated with the rmAb proteins in PBS (0.5 mg/mL) for 2 h at room temperature, 600 rpm. After washing, the wells were incubated with StrepTactin–HRP at 1:1000 and 1:5000 dilutions in MPBST for 1 h at room temperature, 600 rpm. The plate was then washed, and ABTS substrate (Thermo Fisher Scientific, Waltham, MA, USA) was added and incubated in the dark at 37 °C, 600 rpm for 30 min. The absorbance value was read at 405 nm using the SkanIT absorbance reader (Thermo Scientific, Waltham, MA, USA).

A native antigen–antibody ELISA was carried out similarly as described above but with slight modifications, i.e., the coated antigen was 20 µg/mL *T. canis* lysate protein, StrepTactin–HRP conjugate dilution was 1:3000, and the goat anti-rabbit IgG-HRP dilution was 1:5000.

### 2.6. Titration ELISA

Titration ELISA was carried out to determine the binding limit of the rmAb proteins against the rTES-26 antigen. The ELISA was performed as described above, with slight modifications, i.e., rmAb proteins were diluted at concentrations ranging from 500 μg/mL to 7.813 μg/mL, and the dilutions of the StrepTactin-HRP were 1:1000 and 1:3000.

### 2.7. Specificity ELISA

Specificity ELISA was conducted to determine the possible cross-reactivity of the *T. canis* rmAb proteins with non-target antigens. The ELISA was performed as described above by coating 20 μg/mL of non-target recombinant antigens, i.e., *Strongyloides stercoralis* (rNIE and A133), *B. malayi* (B*m*R1 and B*m*SXP) and *T. cati* (rTES-120 cati). StrepTactin–HRP at 1:3000 dilution was used as the detector. The same concentration of rTES-26 was used as the positive control.

### 2.8. Surface Plasmon Resonance Analysis

Surface plasmon resonance (SPR) was performed to characterize the binding affinity and kinetics between the recombinant mAb and the antigen. The CM5 sensor chip surface (Cytiva, Marlborough, MA, USA) was prepared according to standard procedures by immobilizing the rTES-26 protein (antigen) as the ligand on the chip [28] while the mAb solution as the analyte was introduced over the surface of the chip. First, pH scouting of the ligand was performed to make the immobilization more efficient, using 50 μg/mL rTES-26 protein diluted in 10 mM sodium acetate at pH 4.0 to 6.5 (at 0.5 intervals). The carboxylated matrix was activated with an amine coupling approach by injecting a 1:1 ratio of 1-ethyl-3-(3-dimethylaminopropyl)-carbodiimide (EDC) and N-hydroxysuccinimide (NHS). The immobilization of ligand was performed by a second injection of 50 μg/mL rTES-26 protein at the optimal pH on the sensor surface of the BiacoreTM X100 instrument (Biacore, 120 Uppsala, Sweden). Ethanolamine was used in the third injection to block the active surface and then washed to eliminate any residues of the electrostatically attached ligand. The rmAb protein concentrations ranging from 1 μM to 62.5 nM (2-fold serial dilution) were tested against the immobilized ligand with PBS at pH 7.4 as the running buffer. At each cycle, 10 mM glycine–HCl at pH 2.0 was used to unbind any remaining binders and regenerate the active surface. The kinetics and affinity binding of the analyte–ligand during the SPR run were determined using the 1:1 Langmuir and the steady-state models in the Biacore Evaluation Software.

### 2.9. Preliminary Antigen Detection ELISA Using Human Serum Samples

A preliminary antigen detection ELISA was performed using two pooled serum samples from *Toxocara* spp.-seropositive individuals (Sample 1 and Sample 2) and two pooled negative serum samples from healthy individuals (Healthy 1 and Healthy 2). Each pool comprised three anonymized serum samples from our previous serum bank. First, each recombinant monoclonal antibody protein was coated (at 5 μg/mL, 10 μg/mL, and 20 μg/mL) on wells of a Maxisorb ELISA microtiter plate with carbonate buffer (pH 9.6) at 4 °C overnight. The next day, the microtiter plate was washed three times at 5 min intervals with PBST at 800 rpm on a plate shaker to remove unbound antibody protein, then blocked with MPBST for 1 h at 37 °C, 600 rpm. The microtiter plate was washed, and the wells were incubated with serum samples for 2 h at room temperature, 600 rpm. Three serum dilutions were used, i.e., 1:100, 1:200 and 1:300. After a washing step, the wells were incubated with HRP-conjugated secondary antibody for 1 h at room temperature, 600 rpm. Three different conjugates at 1:1000 dilution were used: anti-human IgGF(ab’)2–HRP (Thermo Fisher Scientific, Waltham, MA, USA), anti-human IgGFc–HRP (Invitrogen, Waltham, MA, USA), and anti-human IgG4–HRP (Invitrogen, Waltham, MA, USA). The plate was then washed, and subsequently, ABTS substrate was added and incubated in the dark at 37 °C, 600 rpm for 30 min. The absorbance values were read at 405 nm using the SkanIT reader.

## 3. Results

### 3.1. Recombinant TES-26 Protein Preparation, Verification and Sequence Analysis

The rTES-26 protein was successfully expressed, purified, and verified using SDS PAGE and Western blot. The rTES-26, with a molecular mass size of ~37 kDa, was produced with a good yield and purity suitable for biopanning (Appendix A). The yield of the production of the rTES-26 protein was 2.5 mg/mL of 2 L of bacterial culture.

The BLAST analysis of the rTES-26 antigen sequence against *B. malayi* sequences showed no significant similarity at nucleotide level, and at protein level the percentage of similarities ranged from 28.92% to 40.0%. Appendix A shows the ClustalW alignment of the conserved and non-conserved regions. 

### 3.2. Isolation of Monoclonal Antibodies

At the end of the biopanning process, a polyclonal ELISA was performed and it showed a significant increase in absorbance values from rounds one to three with the OD of 405 nm, ranging from 0.321 to 3.020, indicating enrichments of the rTES-26-specific phage antibodies (Figure 1A). 

Then, a total of 368 antibody clones were screened in the monoclonal ELISA (Figure 1B). Positive clones were selected based on a cut-off OD405 nm value above 0.40 after subtracting the background. Five positive binders were identified, with absorbance values ranging from 0.45 to 3.91. 

### 3.3. Monoclonal Antibody Gene Analysis

The sequences of the five monoclonal antibody clones were analyzed using IMGT/V-QUEST software to determine the identities of the scFv clones based on the human germline sequences. Four clones (22, 48, 49 and 50) showed full-length scFv antibody sequences. However, clone 51 showed a partial scFv sequence with only a light chain having complete complementarity-determining regions (CDR); this clone was excluded from the gene pairing and antibody gene analysis. 

The four scFv antibody clones showed variations in their gene family distributions with three unique gene pairings (Table 1). All the clones showed only functional variable lambda (VL) genes and no variable kappa (VK) gene representation. Additionally, there was a preference in the heavy chain (VH) gene usage, with most clones being derived from VH3 (75%) followed by VH6 (25%). On the other hand, there was an equal gene representation of LV1 (50%) and LV3 (50%) for the light chain. The most common gene pairing was IgHV3–LV1 (50%), followed by IgHV3–LV3 (25%), and IGHV6–LV3 (25%) being equally distributed. The results are depicted in Figure 2 and includes comparison with previously isolated mAbs against other helminth antigens.

### 3.4. Monoclonal Antibody Gene Sequence Analysis

We analyzed the CDR region’s antibody sequence length and amino acid composition since they play a vital role in the binding site topology during antigen–antibody interactions. The range of amino acid (aa) length for the VH was between 7 to 16 aa (data not shown). The HC CDR1 had a distribution of 8 to 9 aa, with 8 aa being the dominant length. CDR2 had 7 to 8 aa, and CDR3 had 15 to 16 aa. The dominant length for CDR2 and CDR3 was 8 aa and 15 aa, respectively. The distribution of the LC CDRs was also varied, with 6 and 8 aa for CDR1 and 3 aa for CDR2. However, the length range for CDR3 was broader, from 11 to 13 aa, with 11 aa as the dominant length (data not shown). 

Amino acid propensity for the enriched scFv CDR regions was observed, and a Pivot chart analysis was performed to study the distribution patterns. Some regions showed random aa distributions, while others showed a skewed aa representation for heavy and light chains (Figure 3). The HC CDR1 showed a higher representation of serine, phenylalanine and asparagine, while HC CDR2 showed a dominance of serine, glycine and isoleucine. The highly diverse HC CDR3 showed an over-representation of serine, leucine, aspartate and phenylalanine. Meanwhile, for LC, the aa of CDR1, CDR2 and CDR3 regions were random but significantly increased in serine. In addition, the LC CDR3 region also showed good frequencies of aspartate, phenylalanine, and valine. 

The aa polarity distribution data of the enriched antibody clone was analyzed (data not shown). All the CDRs in both chains except for CDR-L2 showed similar distribution patterns with higher representation of neutral and small aa. CDR-H1 and CDR-H3 have a similarly high presence of non-polar and relatively large aa. CDR-H3 and CDR-L3 also have a significant non-polar and relatively small aa. Polar and relatively large aa was highly distributed in CDR-L2 and moderately in CDR-L3. In addition, CDR-L3 also showed a good frequency of polar and relatively large aa. All the CDRs in both chains showed the absence of cysteine.

### 3.5. Preparation of Recombinant Monoclonal Antibody Protein

The three clones with unique gene families (clone 48: IgHV3-LV1; clone 49: IgHV3-LV3; clone 50: IgHV6-LV3) were subcloned into pET 51(b) + vector for expression in SHuffle^®^ T7 *Escherichia coli* cells. All clones showed successful insertion of the full-length scFv antibody sequence except clone 50, which showed a mutation that resulted in frameshifts and truncations and was excluded. 

A scaled solubility protein greater than 0.45 on the Protein-Sol server predicts higher solubility than the average soluble *E. coli* protein [29]. The scaled solubility protein values for clones 48 and 49 rmAb proteins were found to be 0.498 and 0.511, respectively (Appendix A). The SDS-PAGE protein profile and Western blot (Figure 4 and Figure 5) showed the expected molecular mass of the rmAb proteins to be approximately 35 kDa. The antibodies were thus successfully expressed and purified at satisfactory levels. The yields of recombinant antibody proteins for clones 48 and 49 were 0.5 mg/mL and 1.5 mg/mL, respectively, for the 2 L culture.

### 3.6. Antigen–Antibody Binding Assays

The antigen–antibody binding was characterized using rTES-26 and *T. canis* native protein in several assays using StrepTactin–HRP to confirm the binding of the rmAb proteins to the target antigens. Two binding assays were performed using rTES-26, i.e., antigen–antibody Western blot and antigen–antibody ELISA (Figure 6A,B). In both assays, the rmAb proteins showed high specific binding to the target antigen, and the ELISA result showed that clone 49 (OD405: 3.3) had a higher absorbance reading than clone 48 (OD405: 1.76). Subsequently, Western blot and ELISA using *T. canis* native antigen were performed, and both results confirmed the binding of the antibody clones to the native antigen (Figure 7A,B). 

### 3.7. Titration ELISA

Titration ELISA was performed to determine the limit of binding of the antibody clones. The range of rmAb protein concentration was 500 μg/mL to 7.81 μg/mL. Clone 49 showed a higher binding strength at lower protein concentration than clone 48 (Figure 8). Clone 49 was able to bind as low as 31.25 μg/mL, while clone 48 showed a binding limit of 62.5 μg/mL. 

### 3.8. Specificity ELISA

The binding of monoclonal antibody proteins was checked against recombinant antigens of other helminths, i.e., *S. stercoralis* (rNIE and rA133), *B. malayi* (rB*m*R1 and rB*m*SXP), and *T. cati* antigen (rTES-120 cati). rTES-26 was used as the positive control. The two clones showed different degrees of binding to the antigens (Figure 9). Clone 48 showed a low level of binding to rB*m*R1, rB*m*SXP and rTES-120 cati antigens and cross-reacted with rNIE and A133 antigens. On the other hand, clone 49 did not cross-react with any of the helminth antigens. Thus, clone 49 showed much higher specificity towards rTES-26 antigen than clone 48.

### 3.9. Surface Plasmon Resonance

Based on a quick immobilization rate (exponential curve) and a high response unit, the pH scouting revealed that pH 5.5 was optimal. Once the sensor surface of the CM5 chip was activated, the rTES-26 ligand was immobilized on the surface. The bindings of clones 48 and 49 rmAb to the immobilized rTES-26 ligand are shown in Figure 10A,B, respectively. 

The SPR data on binding values, association rate constants (ka), dissociation rate constants (kd), and equilibrium dissociation constants (KD) kinetics were studied. Based on the literature, the standard range of the association rate constant ka value is 10^3^ to 10^7^ M^−1^ s^−1^, the dissociation rate constant kd value is 10^−1^ to 10^−6^ s^−1^, and the equilibrium dissociation constants (KD) kinetics and binding values are 10^−3^ to 10^−12^ M [30]. The (KD) kinetics value was obtained by dividing kd with ka. The results of the ka, kd and (KD) kinetics and binding values for the two rmAbs against rTES-26 protein are shown in Figure 10. A smaller (KD) value indicates a higher affinity of the antibody, and it means a stronger binding affinity between the rmAb analyte and the rTES-26 ligand. Both clones showed similar binding strength, i.e., clone 48: 0.02740 and clone 49: 0.0300. Although both clones fell within the typical ranges, clone 48 showed a smaller KD kinetic and binding value, higher association rate, ka, and lower dissociation, kd, constants. The results indicate that clone 48 formed a more stable analyte–ligand complex and showed a stronger binding affinity to rTES-26 than clone 49. 

### 3.10. Preliminary Antigen Detection ELISA Using Human Serum Samples

Antigen detection ELISAs using clones 48 and 49 show a significant difference between pooled *Toxocara* spp.-positive serum and pooled healthy serum using the following parameters: coating concentration of 20 μg/mL, 1:100 serum dilution and anti-human IgG4–HRP (1:1000) as the secondary antibody (Figure 11). The absorbance readings of healthy serum were similar to or higher than the *Toxocara* spp.-positive serum when anti-human IgGF(ab’)2-HRP and anti-human IgGFc-HRP secondary antibodies were used (data not shown). 

## 4. Discussion

Over the past decades, monoclonal antibodies have found tremendous applications in drug targeting, therapeutics and had a significant impact on diagnostics. This is due to their restricted feature that only recognizes a unique antigenic determinant (epitope) on pathogens. Thus, this allows for precise identification of the target organism which is the prime advantage over polyclonal based detection. 

There are a number of studies that have reported the benefit of monoclonal antibodies in the detection of infectious diseases. Among the reported pathogens are *Trichomonas vaginalis* [31], *Leishmania donovani* [32], *Trypanosoma congolense* [33], and *Babesia bovis* [34]. Monoclonal antibody-based systems also have been employed for detection of animal viruses such as bovine herpes virus, cervine herpes virus type I, pseudo rabies virus, calf strain RIT 4237 (sub-group I) and human strain 82–561 (sub group 3) of rotavirus [35,36,37].

Phage display technology has been the most widely used method to isolate monoclonal antibodies [38]. The robustness and high stability of phages have allowed this technology to gain popularity over other isolation technologies. In this study, a previously established helminth antibody phage library was utilized. It was expected to isolate antibodies against proteins from other nematodes since homology among their proteins is expected. The *B. malayi* scFv phage display library was previously used to isolate antibodies against several parasite antigens, i.e., two filarial antigens, B*m*SXP [22] and B*m*R1 [23], *Echinococcus granulosus* antigen B [24], and *S. stercoralis* NIE antigen [25]. In the present study, novel recombinant monoclonal antibodies were isolated against a *T. canis* recombinant antigen using the same library. Initially, BLAST analysis of the rTES-26 antigen sequence was performed against *B. malayi* sequences to determine the extent of sequence identity and whether *B. malayi* express TES-26- like proteins. The results showed that at the nucleotide level, both sequences showed no significant similarity; however, some similarities were seen at the protein level, ranging from 28.92% to 40.0%. The successful isolation of the antibody clones in this study showed that despite not showing high similarities at the protein sequence level, the filariasis immune library could enrich antibodies against *T. canis* protein. 

Among the reported *Toxocara* spp. recombinant proteins, rTES-26 had shown good diagnostic value for detecting *T. canis* infection, hence was used in the present study. A rTES-26 based IgG4 ELISA showed 80% sensitivity (24/30) and 96.2% (204/212) specificity [10]. Another study reported 100% sensitivity (*n* = 6) of a rTES-26 based IgG ELISA [21]. The use of rTES-26 protein has also shown a good accuracy of *Toxocara* spp. diagnosis in other detection platforms such as IgG4-based rapid tests [13] and Luminex bead-based assays [39]. 

In total, four antibody clones comprising three unique gene family-pairing genes (IgHV3-LV1, IgHV3-LV3, and IGHV6-LV3) were successfully isolated. A gene family of previously isolated antibodies against other helminth antigens using the same library was compared. The antibodies against B*m*SXP filarial antigen were mainly derived from functional kappa genes rather than functional lambda genes [22], while antibodies against B*m*R1 filarial antigen showed an equal representation of variable kappa and lambda genes [23]. On the other hand, antibodies to Strongyloides rNIE were dominated by functional variable lambda genes, and only a small percentage of sequences were kappa genes [25]. Similar to rTES-26, antibodies against a hydatid antigen (AgB) were derived from the functional variable lambda gene [24]. An overwhelming predominance of VH3 antibodies can be seen for rTES-26, B*m*SXP, and NIE antibodies, whereas B*m*R1 and AgB antibody clones were mainly represented by VH2 and VH5, respectively. The VL3 antibody is present in the current rTES-26 and all the helminth antigens. The most common gene pairings for the antigens were IgVH3-VK1 for B*m*SXP, IgVH2-VL3 and IgHV2-VK3 for B*m*R1, IgHV3-LV6 for NIE, and IgHV5-LV3 for AgB. They were thus different from rTES-26, whereby VL1 was dominant.

In general, human peripheral blood consists of more kappa than lambda antibodies, with the kappa/lambda ratio being between 1.5 and 2 [40,41]. Nevertheless, this ratio can significantly vary in diseased or antigen-selected populations depending on the class of antibody heavy chain. Other than the present study, the supremacy of lambda subfamilies had also been reported [42,43,44]. For instance, HIV-specific antibodies [45] and antibodies from the mucosal region [40] showed a skewed representation of lambda antibodies. In addition, both kappa and lambda-derived antibodies inherently differ in binding characteristics, as lambda antibodies are more stable when paired with different VH families to produce more stable scFv antibodies due to their higher scFv-pIII fusion protein expression levels. Meanwhile, kappa antibodies showed poorer expression in *E. coli* compared to lambda antibodies [46]. 

The length and the amino acid distribution of the CDR determine the topological variation and information of the binding site motifs during antigen–antibody interactions. Thus, analysis of the sequence length and amino acid composition of the CDR region of the isolated rTES-26 antibody clones was performed. The most focused region is the CDR-H3, which is highly diverse and deemed the B cell fingerprint and its progeny. Although CDR-L3 region variability is less than CDR-H3, it plays a role in the antigen-binding site. The range of CDR length for the HC was 7 to 16 amino acids, and LC was 11 to 13 amino acids. On the other hand, some regions were represented with random aa distributions, while a skewed aa representation for both heavy and light chains was also noted. 

The amino acid usage and polarity of rTES-26 antibody clones against previously isolated antigens are comparable, with some variations. Based on the highly diverse region, HC CDR3 of B*m*R1 antigen was dominated by arginine and alanine [23], while B*m*SXP antigen showed random equal distribution of all aa but slightly higher serine [22]. NIE antigen–antibody clones showed the over-representations of glycine, aspartic acid, valine and asparagine [25]. AgB antigen, represented by only one antibody clone, showed a random aa distribution with the presence of proline, threonine and tyrosine [24]. Meanwhile, rTES-26 antigen–antibody clones showed serine, leucine, aspartate and phenylalanine over-representations. Interestingly, for the light chain of CDR3, despite the presence of other aa in all five antigens, rTES-26 antigen–antibody clones showed the presence of significantly more serine molecules. The polarity distribution of all the five antigens showed a higher representation of neutral and small aa; however, it is not unexpected since antibody clones are rich in aa, such as serine and glycine. 

Following the gene analysis, two clones representing unique gene families (clone 48: IgHV3-LV1; clone 49: IgHV3-LV3) were subcloned into pET 51(b)+ vector for expression in SHuffle^®^ T7 *Escherichia coli* cells. The positive results of the solubility prediction using the Protein-Sol server were consistent with the experimental findings that clones 48 and 49 showed good solubility during expression, with satisfactory yield and purity. 

Expression vectors with dual affinity tags have become increasingly popular since it simplifies purification and enables homogenous preparations of the proteins of interest [47]. In this study, the anti-His tag was used to express and purify the recombinant monoclonal antibody proteins, and the StrepTactin–HRP tag was used for binding assays such as ELISA and Western blot. The binding assays could not be performed without the StrepTactin–HRP tag since both the rTES-26 and rmAb have anti-His tags. The antigen–antibody Western blot and ELISA showed high specificity binding towards recombinant and native forms of *T. canis*. protein. Meanwhile, the titration ELISA showed that clone 49 could bind as low as 31.25 μg/mL while clone 48 had a binding limit of 62.5 μg/mL. 

The recombinant antibody proteins were characterized using specificity ELISA and SPR analysis. Interestingly, clones 48 and 49 showed different degrees of specificity levels against various helminth antigens. Clone 48 showed cross-reactivities against rNIE and A133 antigens and some low-level binding to B*m*R1, B*m*SXP and rTES-120 antigens. Meanwhile, clone 49 exclusively bound to rTES-26 and showed no cross-reactivity against the other antigens tested; thus, may be more useful for diagnostic application. The SPR analysis showed that the clone 48 recombinant antibody protein produced a more stable analyte–ligand complex, indicating a stronger binding affinity to rTES-26 antigen than clone 49. An immune library repertoire contains large amounts of unimmunized antibody clones and biased immunized clones, thus enriched antibodies can have different affinities and specificities. Overall, clone 49 performed better than clone 48 in protein yield and binding affinity.

The preliminary results of the antigen detection ELISA using clones 48 and 49 showed their potential for diagnostic application in detecting *Toxocara* spp. antigen in seropositive human serum samples. However, the ELISA needs to be further optimized and validated with a larger sample size. 

## 5. Conclusions

The present study described the isolation of novel *T. canis*-specific antibodies. Two potential clones were identified, i.e., 48 and 49, and the corresponding recombinant antibodies were produced and characterized. Additionally, this study provides a glimpse into the depth of the antibody repertoire produced from a non-target disease-specific antibody phage display library. Differences and similarities at the antibody gene sequence level provided some insights on *T. canis*-specific antibodies compared to previous helminth antibody clones isolated using the same immune library. In the future, it is essential to elucidate the underlying antigen–antibody interactions by identifying the key binding epitopes of rTES-26 antibody clones through structural analysis using epitope mapping. Further studies are also needed to validate the usefulness of clones 48 and 49, recombinant antibody proteins, as diagnostic reagents to improve the diagnosis of human toxocariasis. The antibody proteins may also be useful in studying host–parasite interactions and therapeutic applications.

## Figures and Tables

**Figure 1 pathogens-11-01232-f001:**
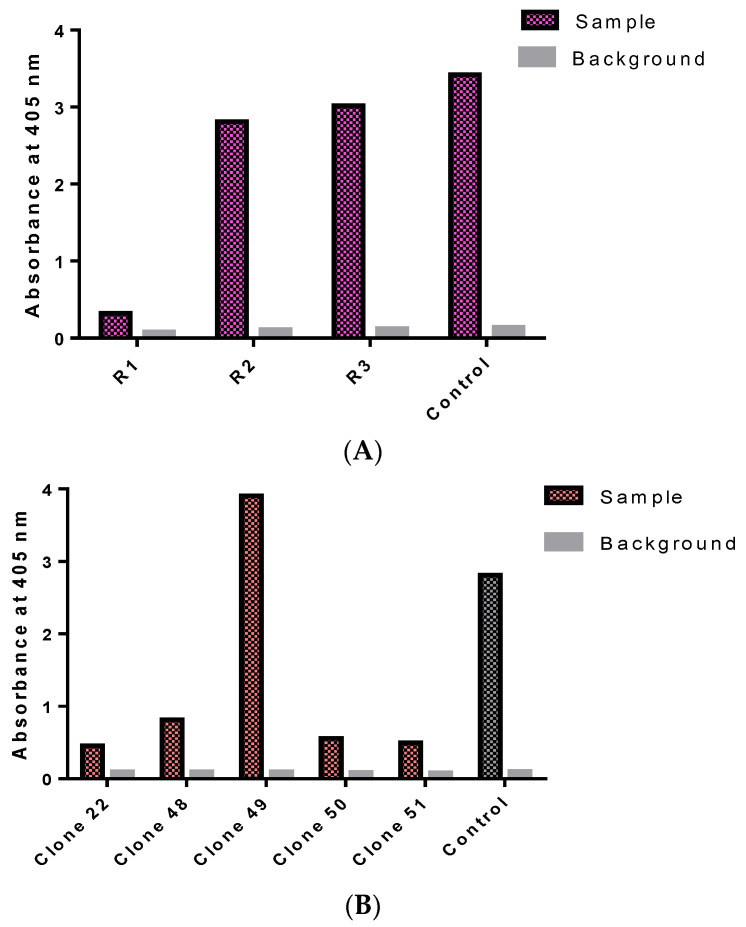
Isolation of rTES-26-specific monoclonal antibodies. (**A**) Polyclonal phage ELISA of rTES-26 antigen during the biopanning cycles (rounds 1–3). (**B**) Monoclonal phage ELISA analysis of scFv clones from the helminth library against rTES-26 antigen.

**Figure 2 pathogens-11-01232-f002:**
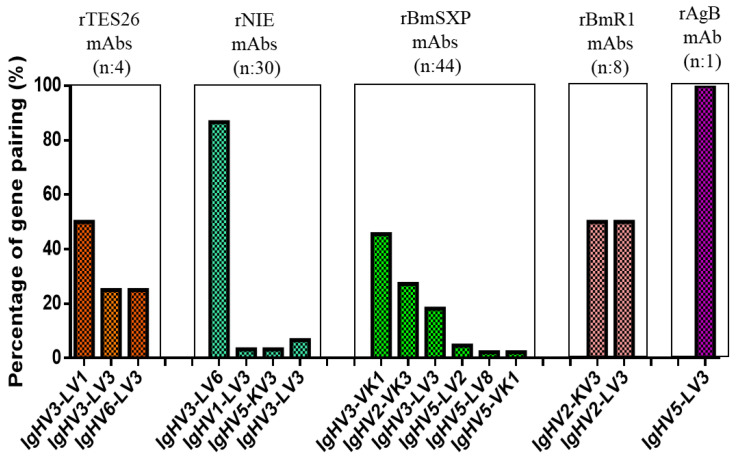
Summary of gene usage of rTES-26-specific antibodies and comparisons with previously isolated helminth protein antibodies.

**Figure 3 pathogens-11-01232-f003:**
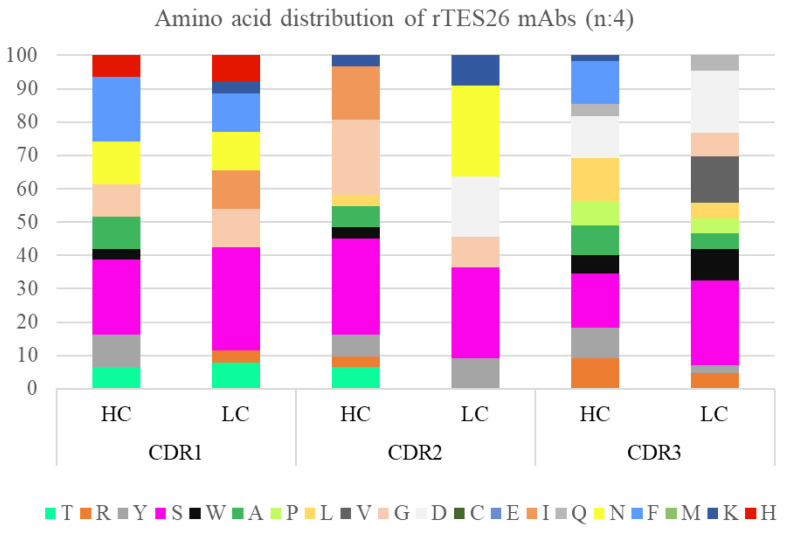
Monoclonal antibody gene sequence analysis of rTES-26-specific antibodies. Amino acid distribution.

**Figure 4 pathogens-11-01232-f004:**
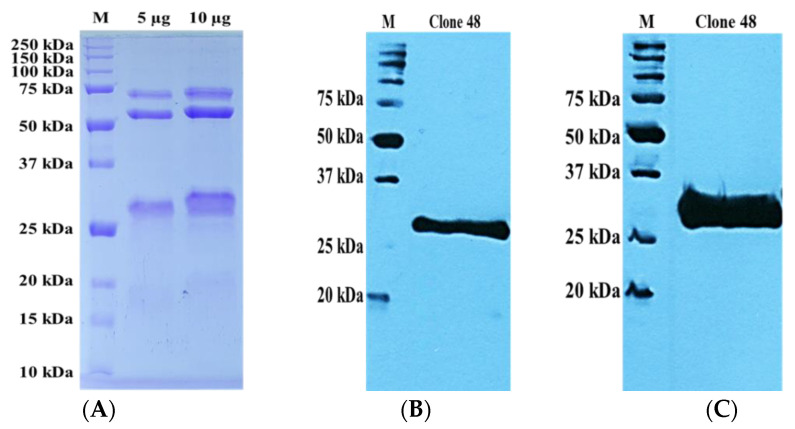
Recombinant monoclonal antibody protein verifications of clone 48. (**A**) SDS-PAGE. (**B**) Western blot analysis using anti-His HRP-conjugated antibody. (**C**) Western blot analysis using StrepTactin–HRP.

**Figure 5 pathogens-11-01232-f005:**
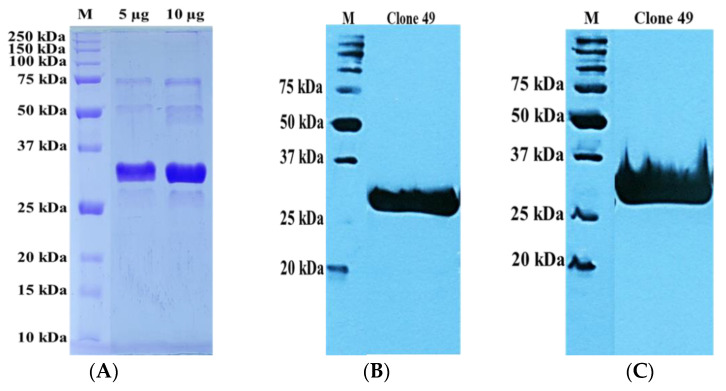
Recombinant monoclonal antibody protein verifications of clone 49. (**A**) SDS-PAGE. (**B**) Western blot analysis using anti-His HRP-onjugated antibody. (**C**) Western blot analysis using StrepTactin–HRP.

**Figure 6 pathogens-11-01232-f006:**
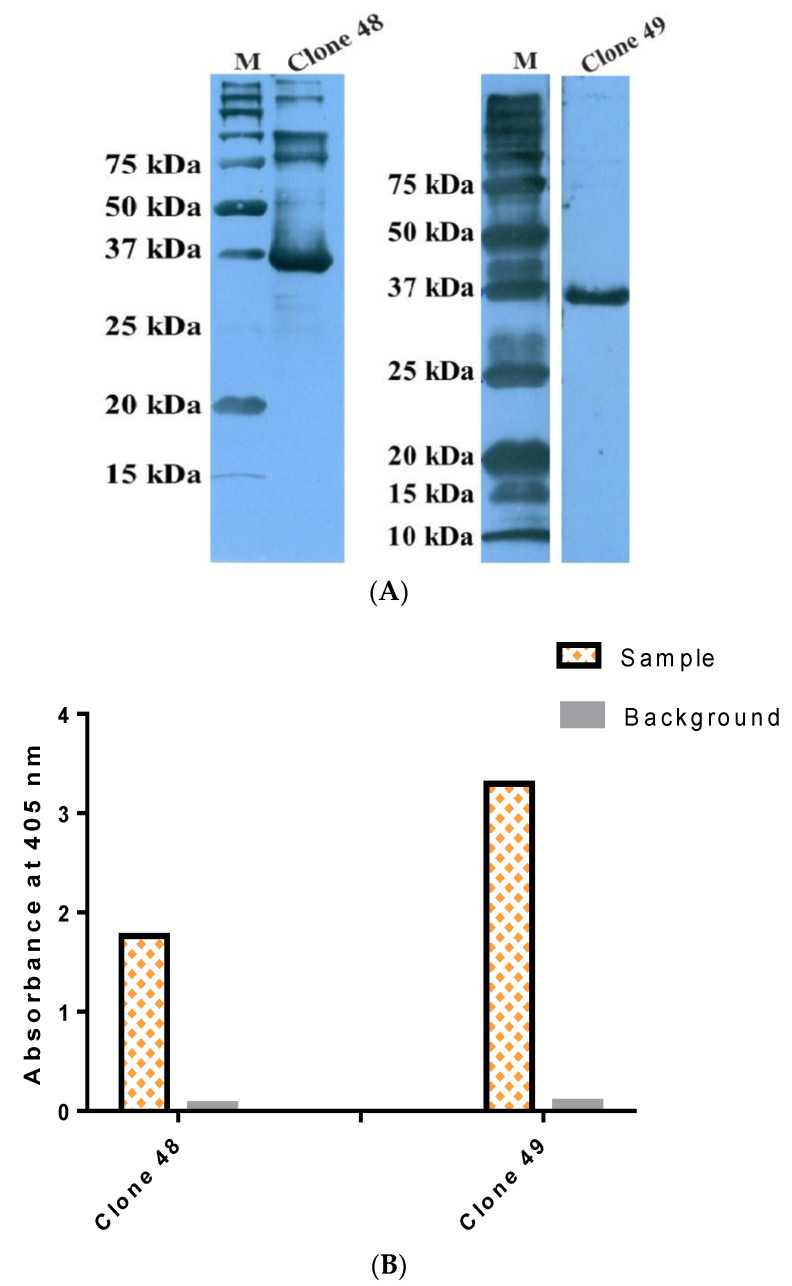
Binding verification of recombinant monoclonal antibody proteins. (**A**) Antigen–antibody Western blot analysis of clones 48 and 49. (**B**) Antigen–antibody ELISA of clones 48 and 49. Both assays were detected using StrepTactin–HRP.

**Figure 7 pathogens-11-01232-f007:**
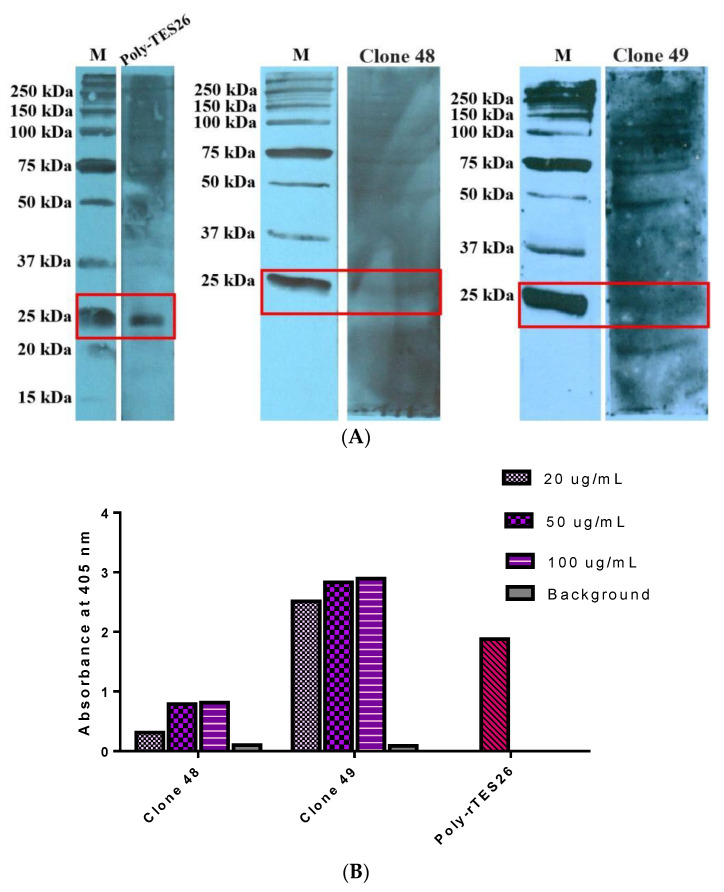
Binding verification of recombinant monoclonal antibody proteins. (**A**) Native antigen–antibody Western blot analysis of clones 48 and 49. (**B**) Native antigen–antibody ELISA of clones 48 and 49. Both assays were detected using StrepTactin–HRP and rabbit anti-rTES-26 polyclonal antibody was used as the positive control.

**Figure 8 pathogens-11-01232-f008:**
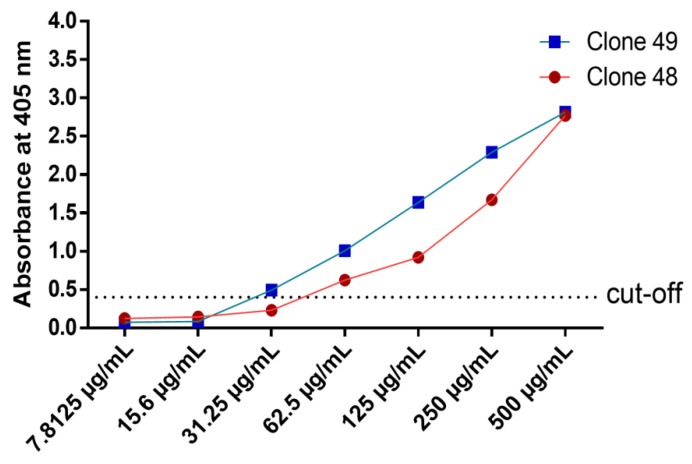
Titration ELISA of antibody clones 48 and 49.

**Figure 9 pathogens-11-01232-f009:**
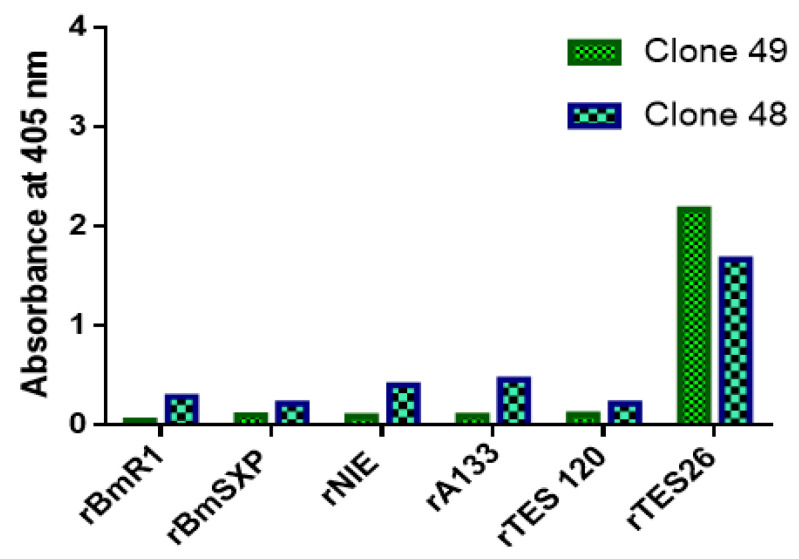
Specificity ELISA of antibody clones 48 and 49 using other helminth antigens.

**Figure 10 pathogens-11-01232-f010:**
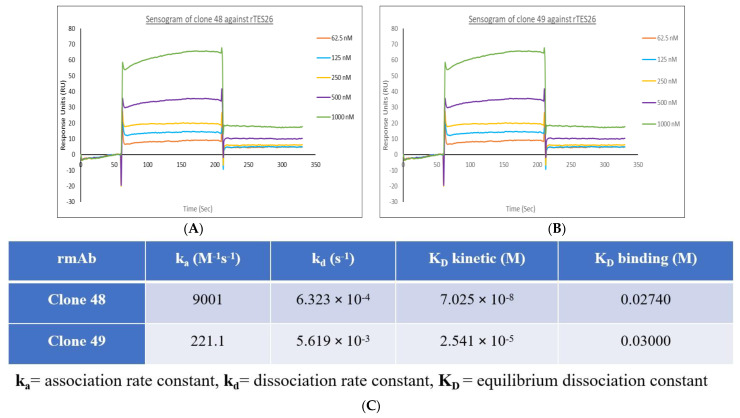
Sensorgrams of different concentrations (1000 nM to 62.5 nM with 2-fold serial dilution) of the two recombinant monoclonal antibodies. (**A**) Clone 48. (**B**) Clone 49. (**C**) Summary of the binding kinetics of clones 48 and 49 against rTES-26 antigen.

**Figure 11 pathogens-11-01232-f011:**
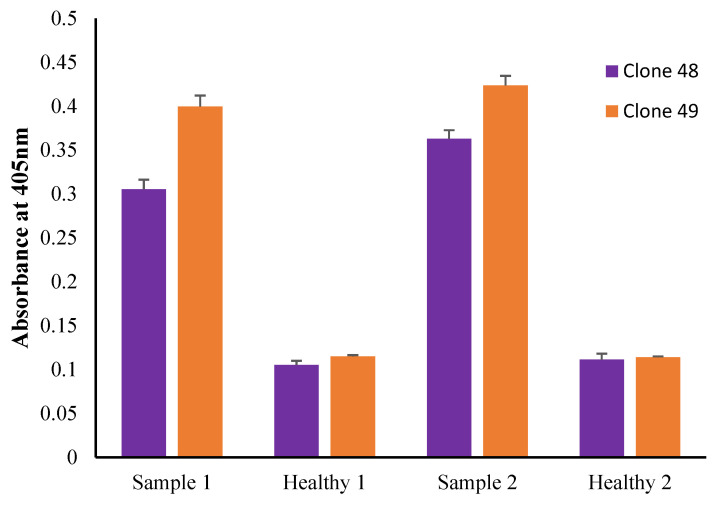
Preliminary antigen detection ELISA using human serum samples. Samples 1 and 2 are pooled serum samples from *Toxocara* spp.-seropositive individuals, and Healthy 1 and 2 are pooled serum samples from healthy individuals.

**Table 1 pathogens-11-01232-t001:** Analysis of gene pairing frequency of isolated antibody clones.

Clone Name	Gene Family	
	Heavy Chain (VH)	Light Chain (VL)
Clone 22	IgVH3	IgVL1
Clone 48	IgVH3	IgVL1
Clone 49	IgVH3	IgVL3
Clone 50	IgVH6	IgVL3
Clone 51	No VH	IgVL3

## Data Availability

Data presented in this study can be found within the Appendix A of this article.

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
