# Peer review of "Isolation and Production of Human Monoclonal Antibody Proteins against a Toxocara canis Excretory–Secretory Recombinant Antigen"

_pathogens, 2022, doi:10.3390/pathogens11111232_

Round 1

Reviewer 1 Report

This manuscript reports the results of an experimental work which resulted in a prototype assay for the serodiagnosis of human toxocariasis. The design of this assay was an immunocapture of circulating soluble antigens from Toxocara spp. larvae using in-house produced monoclonal antibodies. Performances of these monoclonal antibodies were assessed using in-house produced recombinant Toxocara canis soluble larval antigens.

This work represents an excellent piece of applied immunology and biochemistry from a research team that has previously published significant articles pointing out the interest of recombinant antigens for the immunodiagnosis of human toxocariasis.

Whereas the core of this manuscript displays a great value, many shortcomings affect the Introduction, Discussion and References (not appropriate quotations) sections, and give rise to major concerns.

A - Major concerns

1 - The Introduction section is badly constructed and contains too many erroneous statements, an incomplete description of human toxocariasis, not appropriate references, misinterpretations of the content of some references, and so on. .

The authors should shorten the list of references of this section and pay a great attention to the pertinence of the quotations. Examples:

- The authors wrote "The high prevalence of human Toxocara infection is due to dogs being favourite companion animals to humans, a rise in stray dog populations, and the complexity of infection sources [1]"

First, the "rise in stray dogs population" is a factor restricted to some parts of the world, whereas human toxocariasis is present worldwide. Then the reference [1] (Fialho, P.M.M.; Corrêa, C.R.S. A systematic review of toxocariasis: a neglected but high-prevalence disease in Brazil. Am. J. 592 Trop. Med. Hyg 2016, 94, 1193.) does not support this statement, since one can read in this article: ""Accordingly, the specific aim of the present study is to verify, using a systematic review, the study populations and references of existing studies and the prevalence or incidence rates described in each of the investigations".

- Below: "Seroprevalence of Toxocara infections in humans has been estimated at 1.65% in Japan, 2.4% in 37 Denmark, 6.3% in Austria, 7% in Sweden, 14% in the USA, 19.6% in Malaysia, 22% in Iran, 38 81% in Nepal, 85% in Manado (Indonesia), and 87% in the Marshall Islands [4]–[6]"

Sufficient to write that the global seroprevalence of toxocariasis is ...., with variations due to... and to quote a general article, for example "Rostami A et l, Seroprevalence estimates for toxocariasis in people worldwide: A systematic review and meta-analysis, PLOS, 2019". Moreover, the authors will find in this review that human toxocariasis is only slightly more prevalent in children than in adults.

- Below: "Patients with VLM or OLM are often misdiagnosed due to similar signs and symptoms with other helminthic infections, asthma, and allergic reactions"

Erroneous statement concerning ocular toxocariasis.

- Below: "Many Toxocara infections are undiagnosed and untreated due to limitations of existing diagnostic tools". Erroneous statement: current serodiagnostic tools are sufficient for the diagnosis of ACTIVE toxocaral infection, but there is often a lack of awareness concerning human toxocariasis by medical people.

- Below: "Alternatively, an antigen detection assay CAN detect circulating Toxocara antigens"

Since the authors state elsewhere such an assay is not yet available, the use of the conditional tense would be more appropriate.

- Below: "Such an assay is conceivable since the dormant Toxocara larvae 72 in the human body remain metabolically active and able to secrete antigens [17]".

Whether dormant or "hypobiotic" larvae that are arrested in the paratenic hosts' tissues release ES antigens still remains a matter of debate. Moreover the use of ref. 17 [De Savigny, D.H. In vitro maintenance of Toxocara canis larvae and a simple method for the production of Toxocara ES 629 antigen for use in serodiagnostic tests for visceral larva migrans. J. Parasitol 1975, 61, 781–782.] to support this statement is unsound. Toxocara larvae that are maintained in vitro for the production of ES antigens are not dormant, but quite active and motile!

In conclusion, the Introduction should be partially rebuilt, from the beginning to "...vague presentations" according to the following frame: "Human toxocariasis: Definition; Epidemiology (modes of transmission, global prevalence); Pathophysiology; Clinical and Laboratory Pictures (VLM and common/ covert toxocariasis, Ocular Toxocariasis, Neurotoxocariasis). One large review reference is enough for each step. I agree the present manuscript is not the place for a detailed recall about human toxocariasis. Therefore I suggest to the authors to carefully read 3 or 4 excellent reviews on this topic, to make a short synthesis and finally to have the text read by a specialist in Parasitology from their University

2 - Material and methods/ 3.10 Preliminary antigen detection ....

This prototype immunocapture assays has been designed using a recombinant TES-26 antigen as "target". This situation is different from the "real life", where the assay will have to capture a native 26 kDa fraction which moreover will be mixed with other fractions of the ES Ag complex.

In order to check the ability of the prototype to work in real conditions, the authors used 2 batches of pooled sera, negative and positive. This was a correct but insufficient approach, since the concentration of native ES Ag in the positive pooled batch was unknown.

Why the authors did not test increasing concentrations of native ES antigen diluted in negative human sera? This would have been an excellent simulation of the real diagnostic conditions. The authors should give accurate explanations for their choice.

3 - Discussion

The text from "Toxocariasis is a silent public health issue..." to "...address challenges related to antibody assays" belongs to the Introduction section and should be moved there.

Moreover, the rest of the Discussion section is rather a recap of the operational bench procedure than a real discussion. Therefore the Discussion section should be rebuilt and drastically improved.

B - Minor concerns

1 - The Material and Methods section is very dense and detailed, and quite represents a bench manual. Perhaps this section could be reduced.

2 - Are all the figures necessary? At least, the "solar wheel" in Fig. 3 should be deleted.

3 - In Fig. 8, the concentrations should be written " .... µg/ mL", not ".... ug/ml"

Reviewer 2 Report

This paper described the isolation of two scFvs against Toxocara canis recombinant TES-26 antigen (rTES-26) from a phage display library, their purification, and characterization.

This study is interesting because antibodies against T. canis are required to develop diagnostic tools to detect this pathogen. The study was done well; however, from my point of view, some minor issues need to be explained and resolved.

My comments are

-  Lines 112-114  The SDS-PAGE gel of rTES-26 antigen from your supplementary material shows some bands (around 75-100kDa). Are those bands contaminating proteins from the bacterial culture? How do we know that we are selecting scFv clones against rTES-26 and not the proteins from those bands?

- Line 374 “Western blot analysis of clones 48 and 49” These membranes were probed with scFv clones and then the scFv clones were detected with Strep Tacting-HRP. The scFv clone 48 detected several bands in addition to the 37 kDa rTES-26 band. Clone 49 detected only a single band. Is clone 48 less specific than clone 49? What is the specificity of these scFvs? Did you try testing the binding of scFvs to different unrelated proteins to compare it with their binding to TES-26?

- Lines 380 “Native antigen-antibody Western blot analysis of clones 48 and 49,..” Figure 7 A shows 3 membranes where the native form of TEST 26 is detected with polyclonal antibodies, scFv clone 48, and ScFv clone 49. After that, the scFvs were detected with Strep Tactin HRP.  Unfortunately, there were too many bands detected in the membrane incubated with clones 48 and 49, but even with the polyclonal antibody. Could you please try improving this Western blot? Maybe the secondary antibodies are giving too much background. Please, test the secondary antibodies alone without primary antibodies to see if the secondary antibodies are giving you background. Please, include a line with an unrelated protein as a negative control. I know that Western blot using scFv is tricky. This can be sometimes solved by labeling the scFv with HRP if that is possible.

Round 2

Reviewer 1 Report

A - Follow the international standards when writing the names of the parasites:

- write the names in italic

- at the first appearance in the write, the name should be given in full: "Toxocara canis", or "Toxocara cati", and so on.

- then the genus name should be abbreviated: "T. canis", etc.

- when all the Toxocara species are considered, write "Toxocara spp."

Check and correct throughout the manuscript

B - When nouns are enumerated, they should be sorted in alphabetical order, or in logical order, or in a combination of both.

Examples:

- write: "Clinical forms of toxocariasis include visceral larval migrans (VLM), covert or common toxocariasis (CT), ocular larval migrans (OLM) and neurotoxocariasis (NT)"

Logical order: VLM and CT are generalized forms, major for VLM, minor for CT. OT and NT are compartmentalized forms, the most frequent being OT.

- write: "...to different sites such eyes, liver, lungs, and central nervous system".

Alphabetical order

Check and correct throughout the manuscript

C - The circos figure has not been deleted p. 13.

D - Delete this part of sentence "which may suffer from cross-reactivity with antibodies to other parasites" because it is redundant with the previous sentence: "Another major drawback of native TES antigens is the potential cross-reactions with other soil-transmitted helminths [10]."

Move ref. #14 close by ref. #15. 

E - ref. #4 not called in the text
